# The role of conjunctive representations in prioritizing and selecting planned actions

**Atsushi Kikumoto[1,2]\*, Ulrich Mayr[3], David Badre[1,4]**

[1]Department of Cognitive, Linguistic, and Psychological Sciences, Brown University, Providence, United States; [2]RIKEN Center for Brain Science, Wako, Japan; [3]Department of Psychology, University of Oregon, Eugene, United States; [4]Carney Institute for Brain Science, Brown University, Providence, United States

**Abstract** For flexible goal-directed behavior, prioritizing and selecting a specific action among multiple candidates are often important. Working memory has long been assumed to play a role in prioritization and planning, while bridging cross-temporal contingencies during action selection. However, studies of working memory have mostly focused on memory for single components of an action plan, such as a rule or a stimulus, rather than management of all of these elements during planning. Therefore, it is not known how post-encoding prioritization and selection operate on the entire profile of representations for prospective actions. Here, we assessed how such control processes unfold over action representations, highlighting the role of conjunctive representations that nonlinearly integrate task-relevant features during maintenance and prioritization of action plans. For each trial, participants prepared two independent rule-based actions simultaneously, then they were retro-cued to select one as their response. Prior to the start of the trial, one rule-based action was randomly assigned to be high priority by cueing that it was more likely to be tested. We found that both full action plans were maintained as conjunctive representations during action preparation, regardless of priority. However, during output selection, the conjunctive representation of the high-priority action plan was more enhanced and readily selected as an output. Furthermore, the strength of the high-priority conjunctive representation was associated with behavioral interference when the low-priority action was tested. Thus, multiple alternate upcoming actions were maintained as integrated representations and served as the target of post-encoding attentional selection mechanisms to prioritize and select an action from within working memory.

**\*For correspondence:**
atsushi_kikumoto@brown.edu

## Editor's evaluation

This study presents important findings regarding the control processes that underlie planned actions. The evidence for the authors' conclusions is compelling, using state-of-the-art model-based decoding analysis. This work is of interest to cognitive scientists spanning the subfields of cognitive control, working memory, and motor control.

## Introduction

Guiding actions in a complex environment often involves specification and selection of multiple candidate plans (*Cisek, 2007*). Our ability to prepare and then single-out a planned action allows us to keep actions for several potential scenarios ready, and then decide which to execute based on their adaptive utility in the current context. Such flexible action control requires separate mechanisms for rapidly translating incoming sensory information into action-oriented plans in a context-dependent manner and for deciding to enact those plans as behavioral output (*Cisek and Kalaska, 2005*; *Gallivan et al., 2015*). To this end, working memory is thought of as a core interface, both for assembling and

maintaining task-relevant plans, as well as gating a subset of this information to use when actions become relevant (*Chatham and Badre, 2015*; *Myers et al., 2017*; *Kriete and Noelle, 2011*; *Frank and Badre, 2012*; *Badre and Frank, 2012*).

Top-down prioritization and selection of the encoded information have been commonly recognized as a function of working memory (*Souza and Oberauer, 2016*; *Gazzaley and Nobre, 2012*). Retrospectively directing attention to specific memories (e.g. via retro-cues) makes the targeted content more accessible for later retrieval and more resilient against interference, while increasing its neural decodability (*LaRocque et al., 2013*; *Yu et al., 2020*; *Ester et al., 2018*; *Levin et al., 2022*). However, although the primary function of working memory is to guide future actions rather than passively store past sensations (*Miller et al., 2018*; *Olivers and Roelfsema, 2020*; *Nobre and Stokes, 2019*), control over working memory has been studied mostly with regard to sensory representations (e.g. words, letters, symbols, faces, scenes, and so forth) that are independent from their prospective goals and actions (*Myers et al., 2017*; *Heuer et al., 2020*; *van Ede, 2020*). Because realistic goal-directed actions are supported by multiple task features, which often influence one another dynamically (e.g., a hierarchical gating; see *Rac-Lubashevsky and Frank, 2021*; *Ranti et al., 2015*; *Frank and Badre, 2012*), it remains an open question how the full complement of task-relevant features comprising an action plan is managed together within working memory.

One important representational format that may support control over planned actions is a conjunctive representation that evolves over the course of action planning (*Hommel, 2019*; *Frings et al., 2020*). This conjunctive representation, also termed an event or task file, binds critical task-relevant features —including not only the sensory representation (i.e. stimulus), but also the action rule (i.e. context) and response—into an integrated representation that uniquely couples sensory and motor information for a specific goal (*Mayr and Bryck, 2005*). Several theories of cognitive control proposed that over the course of action planning, the formation of such an integrated representation is essential for an action to be executed (*Frings et al., 2020*; *Henson et al., 2014*; *Hommel, 2004*; *Logan, 1989*). Recent studies that applied a decoding analysis to the distributed pattern of EEG activity revealed that humans form such a conjunctive representation while preparing context-dependent actions (*Kikumoto and Mayr, 2020*; *Takacs et al., 2021*). The temporal trajectories of conjunctive representations were dissociable from other action representations, and their trial-to-trial fluctuation in strength strongly correlated with efficient action control (e.g. stopping of actions or transitions of action over trials) over and above other task-relevant representations (*Kikumoto and Mayr, 2020*; *Rangel et al., 2022*).

Because a conjunctive representation uniquely specifies the to-be-executed action, it may be an efficient, output-oriented format for maintenance, prioritization, and retrospective selection of the planned actions for future behavior (*Badre et al., 2021*). Yet, previous studies have focused on a single episode of action selection. Thus, it is unknown whether more than one competing conjunctive representation can be maintained in parallel. If multiple conjunctive representations could be prepared concurrently, action prioritization may primarily modulate output gating of prepared action plans. In contrast, if there is a capacity limit on holding multiple event-file-like representations, maintenance of a given action plan should be compromised by trying to simultaneously prepare for alternative action plans.

In the current study, we investigated the role of conjunctive representations in the maintenance and prioritization of multiple candidate action plans for an upcoming response. Participants prepared two independent actions with unique stimulus-response mappings and chose one of them as a final response based on a retrospective cue. One of the actions was, on average, more likely to be tested, encouraging selective prioritization of that action until output selection.

We hypothesized that (a) future actions are maintained concurrently in the form of multiple conjunctive representations during action preparation and (b) conjunctions are the primary target of prioritization and selection once a planned action is selected (over and above individual components of the action plan like the stimulus, rule, or response). The conjunction representational similarity analysis (RSA) model makes a unique prediction of the pattern of similarity over and above the patterns expected by other constituent action features such as rules, stimuli, and responses (*Figure 1c*), and it tests whether a specific instance of action is distinct from other S-R mappings (*Figure 1b*).

To preview our results, response times (RTs) and errors indicated that participants prepared both actions but held the higher-priority action in a privileged state. By applying a time-resolved RSA to

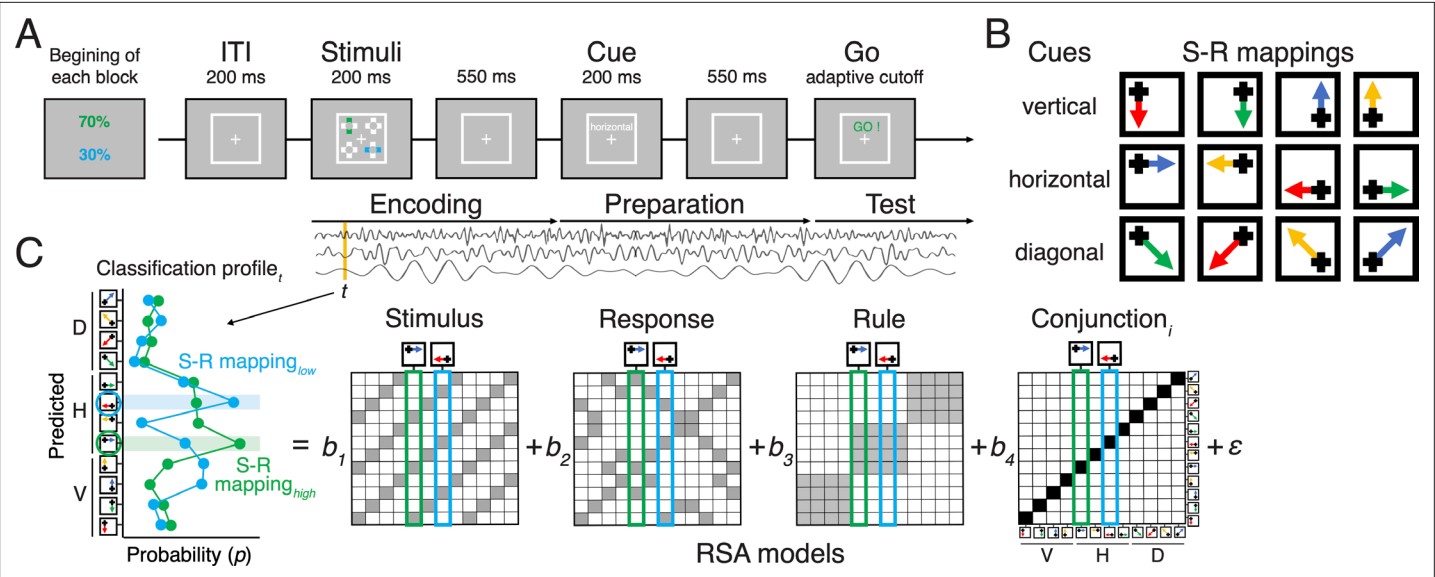

**Figure 1.** Task design and the procedure of decoding analysis. (**A**) Sequence of trial events in the rule-selection task with two independent action plans. Test probabilities of each action are assigned randomly every block. (**B**) Spatial translation of different rules (rows) mapping different stimuli (columns) to responses (arrows). (**C**) Schematic of the steps used for representational similarity analysis (RSA). For each sample time (t), a scalp-distributed pattern of EEG power was used to decode the specific rule/stimulus/response configuration of two actions of a given trial. The decoder produced sets of classification probabilities for each of the possible action constellations. The profile of classification probabilities reflects the similarity structure of the underlying representations, where action constellations with shared features are more likely to be confused. The figure shows an example of classification probabilities for two actions cued by a shared rule and two independent stimuli. For each trial and timepoint, the profile of classification probabilities was regressed onto model vectors as predictors that reflect the different, possible representations. In each model matrix, the shading of squares indicates the theoretically predicted classification probabilities (darker shading means higher probabilities) in all possible pairs of constellation. The coefficients associated with each predictor (i.e. *t*-values) reflect the unique variance explained by each of the constituent features and their conjunction for each action plan.

EEG, we found that conjunctive representations of both high- and low-priority actions were maintained up until the moment of selection. Furthermore, the conjunctive representation of the high-priority action was more enhanced and readily selected as an output, and it produced interference when the low-priority action was selected instead. Modulation based on priority was weak for stimulus representations, suggesting that the conjunction is a primary target of prioritization and selection. Thus, consistent with our hypothesis, multiple candidate actions can be maintained, prioritized, and selected as integrated, but independent action representations.

## Results
### Behavior

The pattern of behavioral results confirmed that the high-priority action led to more efficient action selection (*Figure 2*). When the high-priority action was tested as opposed to the low-priority action, participants responded significantly faster, $F_{(1,23)} = 40.31$, MSE = 962, p<0.001, $M=371$ ms, SE = 8.7 ms for high priority and $M=428$ ms, SE = 8.7 ms for low priority, and produced fewer errors, $F_{(1,23)} = 31.81$, $MSE <.001$, p<0.001, $M=.11$, SE = .01 for high priority and $M=.22$, SE = .01 for low priority. Likewise, a trial-wise adaptive cutoff (i.e. response deadline) was significantly shorter for high-priority action, $F_{(1,23)} = 33.72$, MSE = 7479, p<0.001, $M=823$ ms, SE = 13 ms for high priority and $M=968$ ms, SE = 13ms for low priority.

Most errors were responses corresponding to the untested action plan rather than random responses. This suggests that participants encoded, and likely maintained, both action plans until the test period but then occasionally selected the wrong plan. Such swapping errors occurred at significantly higher probability when the low-priority action was tested, again consistent with participants prioritizing the high-priority action during maintenance, $F_{(1,23)} = 12.59$, $MSE <.001$, p=0.002, $M=.74$, SE = .01 for high priority and $M=.81$, SE = .01 for low priority. Therefore, the behavioral results

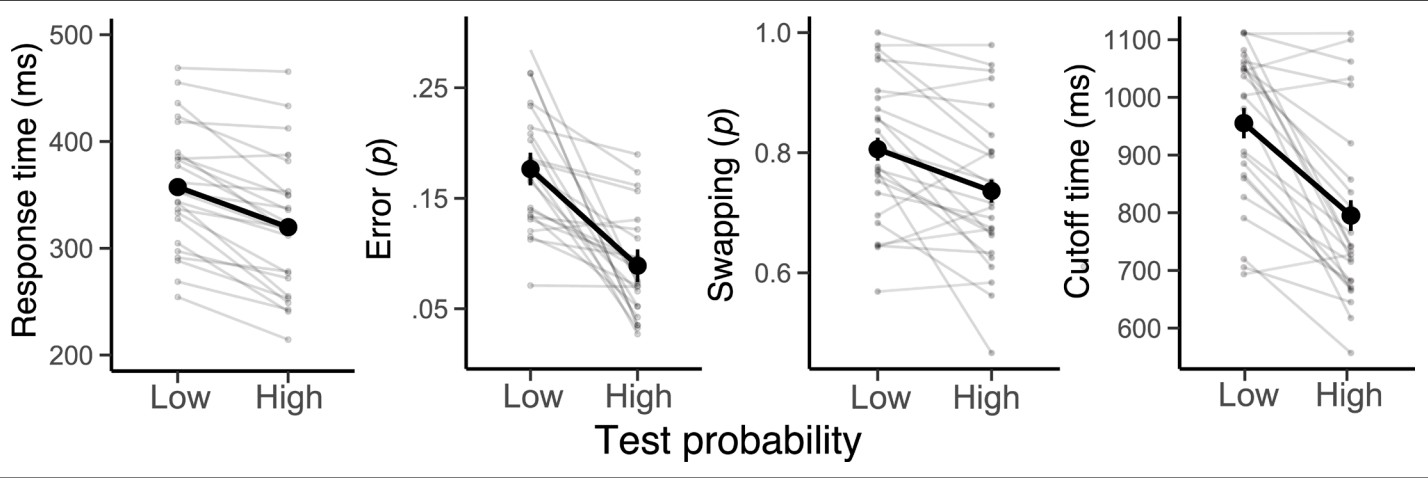

**Figure 2.** The effect of priority of actions on behavior. Average response times, error probability, swapping probability, and cutoff time (i.e. trial-to-trial response deadlines) in the low and high test probability conditions. Note that a swapping error was computed out of all committed errors, rather than all trials. Error bars specify 95% within-subject confidence intervals. Faint lines plot individual participants.

indicate that both of the required actions were prepared, yet the action associated with the high test probability was prioritized over the other action.

## Action representations during early encoding and preparation phase

To assess how two planned actions are maintained and selected via retro-cues, we decoded the representations of action features (i.e. stimuli, rules, responses, and conjunctions for each action) from the time-resolved patterns of EEG activity using RSA. Then, we used a mixed-effect model to assess the effect of priority on the quality of representations and subsequent selection performance at the level of single trials.

The RSA revealed unique temporal trajectories for the individual features of high- and low-priority action plans (*Figure 3*). During the encoding phase, a mixed-effect model revealed that representations of two stimuli for each action were encoded, $t(1,23) = 7.45$, beta = .183, 95% CI [.136.230] for high priority; $t(1,23) = 6.92$, beta = .168, 95% CI [.121.216] for low priority; and they quickly decayed yet remained active during the preparation phase, $t(1,23) = 3.65$, beta = .036, 95% CI [.016.052] for high priority; $t(1,23) = 3.20$, beta = .017, 95% CI [.006.027] for low priority. During the preparation phase, the cued rule context shared by both plans was specified and maintained robustly, $t(1,23) = 6.59$, beta = .144, 95% CI [.101.186]. Furthermore, distinguishable conjunctive representations for both required actions emerged, $t(1,23) = 2.61$, beta = .142, 95% CI [.004.025] for high priority; $t(1,23) = 2.10$, beta = .008, 95% CI [0.001.016] for low priority. In contrast, the information about required responses was not reliably decodable during the preparation phase, $t(1,23) = -.514$, beta = .002, 95% CI [–.010.006] for high priority; $t(1,23) = -1.57$, beta = –.008, 95% CI [–.017.002] for low priority.

Before the test phase, none of these representations is significantly modulated by different test probability values, $t(1,23) <1.62$, suggesting that neither action plan was selectively prioritized during preparation. Nonetheless, these results indicate early encoding and maintenance of conjunctive representations during the preparation phase and preceding the response. Note that in principle, candidate motor responses could have been fully prepared. Yet, action plans were maintained as conjunctive representations instead of response representations during the preparatory phase.

## Output gating of action representations during the test phase

In the test phase, participants were explicitly cued which prepared actions to select and made their response prior to the deadline (*Figure 1A*). We hypothesized that prioritization would modulate the states of selected and non-selected action representations. Specifically, we predicted that when a prepared conjunction is prioritized, this should facilitate selection of that plan, whereas a high-priority plan that is not selected should interfere with the execution of the one that is selected.

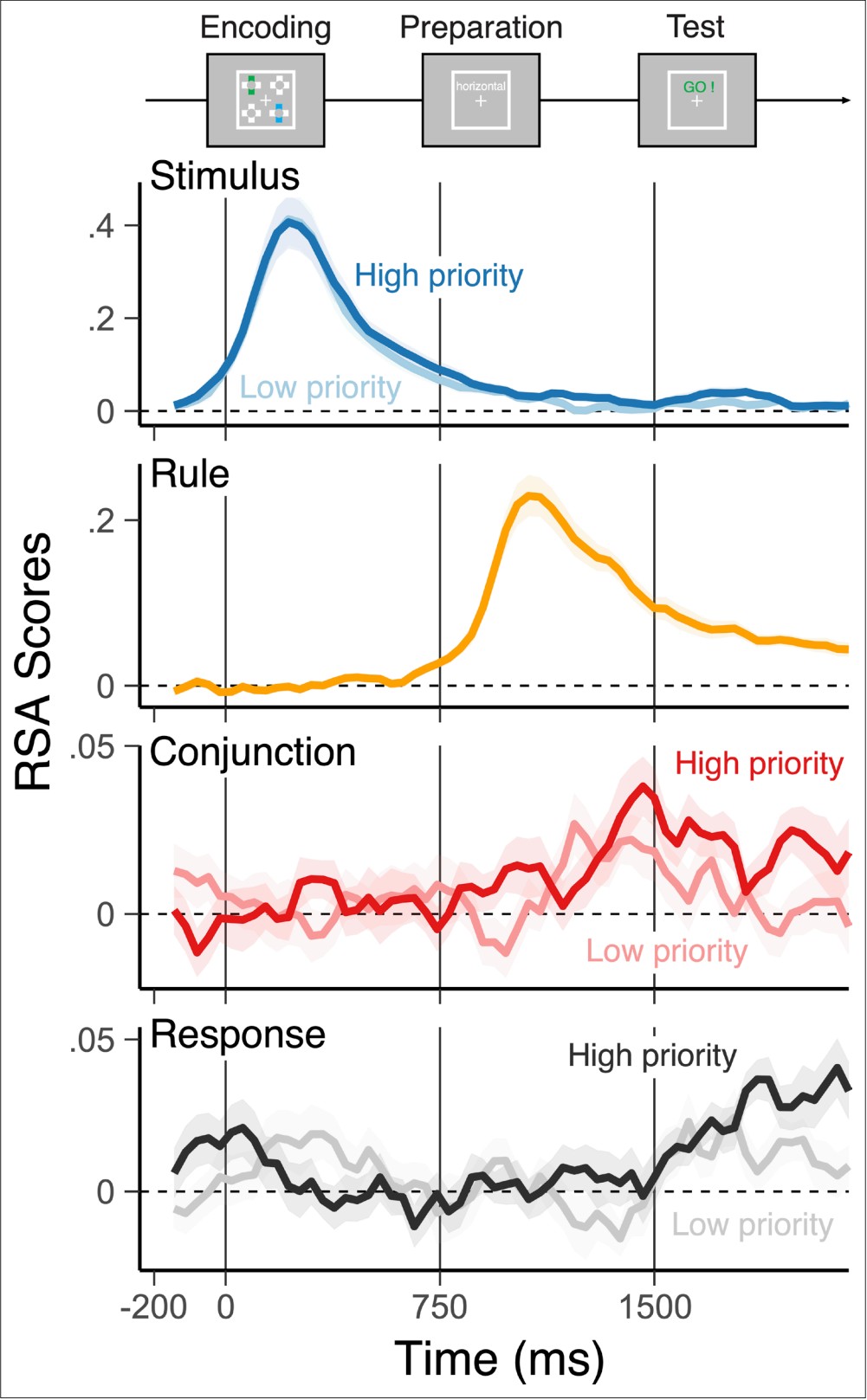

**Figure 3.** Trajectories of decoded representations of two actions in the different levels of priority. Average, single-trial *t*-values associated with each of the basic features (rule, stimulus, and response) and their conjunction derived from the representational similarity analysis (RSA), separately for high-priority (bold colors) action and low-priority action (faint colors). Shaded regions specify the 95% within-subject standard errors.

*Figure 3 continued on next page*

*Figure 3 continued*

The online version of this article includes the following figure supplement(s) for figure 3:

**Figure supplement 1.** Decoded representations of two actions using frequency-specific scalp topography.

Although, on average, representations of individual features of high- and low-priority actions mostly became decodable only when they were cued as an output (*Table 1*), conjunctive representations were significantly modulated by the selection demand (i.e. cued as an output or not) dependent on their priority (*Figure 4*; see *Table 2* for the main effects). Specifically, high-priority items were more active than low-priority items when they were cued, but the reverse was the case when low-priority items were cued.

Notably, the pattern of results in *Figure 4* seems to indicate that the interaction for the conjunctive representation arises later than the representation for the response, particularly when the low-priority item is tested. This would be unexpected if the conjunctive representation informs encoding and selection of the response. Thus, we performed exploratory analyses to test the timing of the interaction effect.

First, we tested if there are significant dependencies of the time interval and the type of representation (conjunction vs. response) on the interaction effect (the priority by the test type). We found that this four-way interaction (priority × test type × representation type [conjunction vs. response], × time interval [1500–1850 ms vs. 1850–2100 ms]) is not significant for both stimulus-aligned data, $t(1,23)$ = 1.65, beta = .058, 95% CI [–.012.015] and response-aligned data, $t(1,23)$ = –1.24, beta = –.046, 95% CI [–.128.028]. These observations were not dependent on the choice of time interval, as other time intervals were also not significant. Therefore, we cannot rule out the possibility that this pattern is due to differences in noise over the interval, and so we cannot draw strong conclusions about the fine-grained temporal effects between conditions within the pre-specified trial phases (i.e. encoding, maintenance, selection).

This caveat notwithstanding, however, further exploratory analysis suggests that the late-emerging interaction in the low-priority test condition was more likely due to a slow decline in the strength of the untested high-priority conjunction rather than late selection of the low-priority conjunction (*Figure 4—figure supplement 1*). In particular, the tested low-priority conjunction emerged early and was sustained when it was the tested action and declined when it was untested (–226−86 ms relative to the response onset, cluster-forming threshold, p<0.05). This decline eventually resulted in a significant difference in strength between the tested versus untested low-priority conjunctions prior to the commission of the response. Importantly, the high-priority conjunction also remained active in its untested condition, declining even later. Indeed, it does not decline significantly relative to trials when it is the tested action until after the response is emitted. It is this slow decline of the unselected conjunction that resulted in the late emerging interaction of priority and test type, rather than a slow emergence of the selected one.

**Table 1.** Trial-by-trial representational similarity analysis scores of high- and low-priority actions in each test probability context during the test phase.

| Decoded features | Tested high priority | | Tested low priority | |
|---|---|---|---|---|
| | *beta* (CI) | *t*-Value | *beta* (CI) | *t*-Value |
| Rule | 0.071 (.05.09) | 6.08*** | 0.069 (.04.09) | 4.76** |
| Stimulus (high priority) | 0.034 (.01.05) | 3.41* | 0.021 (.01.03) | 2.9* |
| Conjunction (high priority) | 0.021 (.01.04) | 3.57* | 0.007 (–.01.02) | 0.86 |
| Response (high priority) | 0.034 (.02.05) | 5.04** | 0.009 (–.01.03) | 1.04 |
| Stimulus (low priority) | 0.011 (.01.02) | 2.20* | 0.014 (–.01.03) | 1.30 |
| Conjunction (low priority) | 0.003 (–.01.01) | 0.61 | 0.001(–.01.02) | 0.12 |
| Response (low priority) | 0.008 (–.01.02) | 1.61 | 0.015(–.01.03) | 1.82 |

Note: *, **, ***, and a dot indicate p<0.05, p<0.01, p<.001.

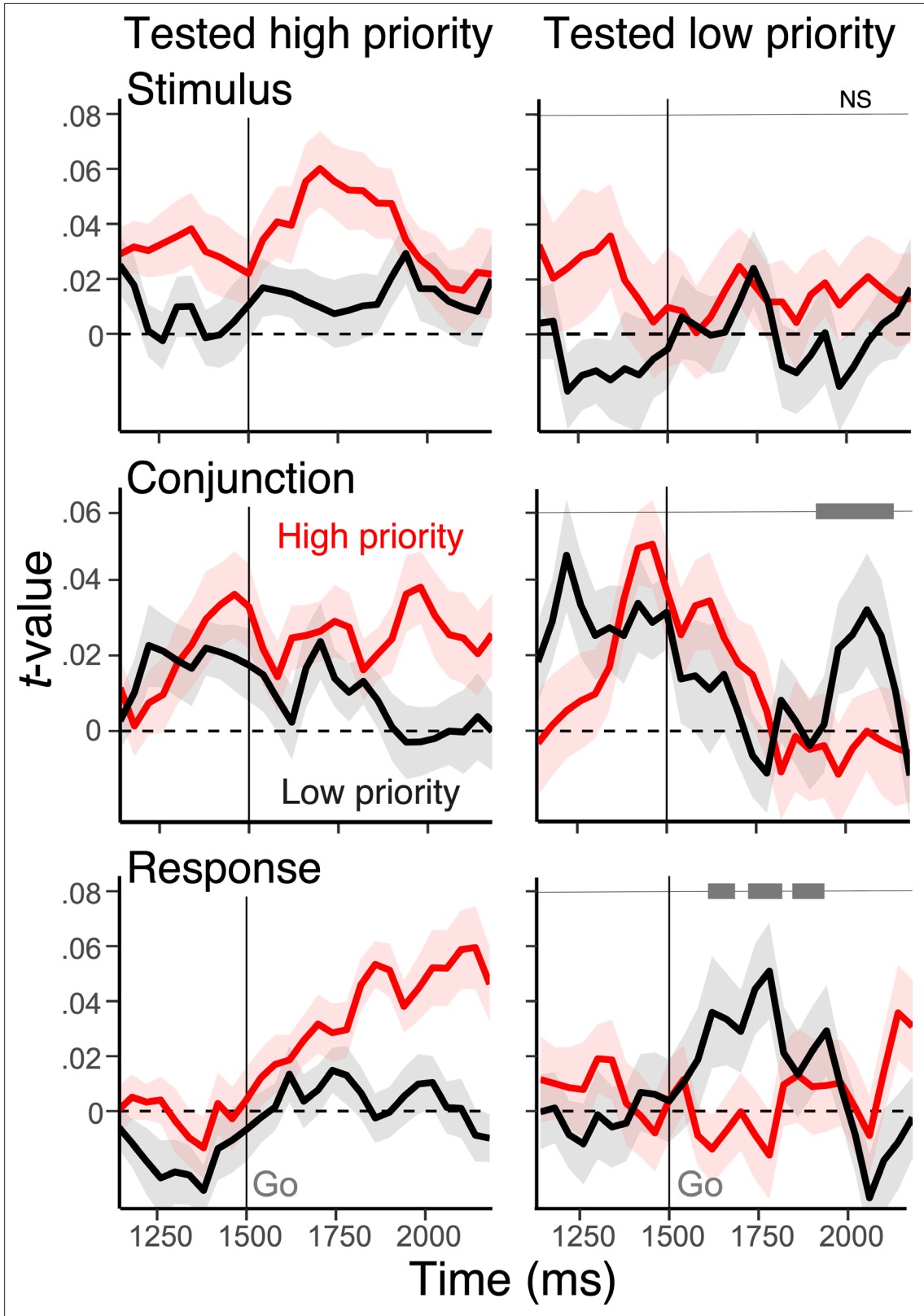

**Figure 4.** Modulation of action representations during selection using stimulus-aligned EEG. Average, single-trial *t*-values associated with the stimulus, conjunction, and response derived from the representational similarity analysis (RSA), separately for high-priority (red) action and low-priority action (black). The left panels show RSA scores when the high-priority action was tested, whereas the right panels show the result for the low-priority action.

*Figure 4 continued on next page*

*Figure 4 continued*

Shaded regions specify the within-subject standard errors. On the right side panels, the gray bars at the top show clusters with a significant interaction effect between the priority and the test type of actions after cluster-based correction (cluster-forming threshold, p<0.05).

The online version of this article includes the following figure supplement(s) for figure 4:

**Figure supplement 1.** Modulation of action representations during selection using response-aligned EEG.

We next tested whether there was a relationship in the strength of the high- and low-priority items with each other. We found that the strength of high- and low-priority conjunctions did not correlate with each other over trials across during the entire preparation and selection period, $t(1,23) > -1.40$ for all individual time samples. The same effect was observed when we focus on trials where we observed strong evidence of both conjunctions, $t(1,23) > -1.72$. Although one cannot draw a strong conclusion from the null effect (i.e. no correlations), this result is consistent with the idea that cross-over pattern of the results was not driven by direct competition among high- and low-priority conjunctions.

Finally, a similar two-way interaction of priority and test condition was observed for response representations but not for the stimulus representations (*Figure 4*).

In summary, then, these results indicate that not every action feature is boosted by the retro-cue, and the conjunction is one of the targets of attentional modulation for selective output gating. Furthermore, our results suggest that the two conjunctions can be maintained in a somewhat parallel manner, and prioritization modulates output gating, not necessarily preparatory activation.

We next considered the effect of the strength of the conjunctive representations on behavior. We observed an asymmetric interaction of action representations with different levels of priority on trial-to-trial response times and accuracies depending on the test type (*Figure 5*). Specifically, stronger representations of a stimulus, response, and conjunction of the high-priority action predicted faster and more accurate responses when that high-priority action was tested by the cue. Such facilitatory effects were diminished or reversed into interference, when the low-priority item was tested as a function of the strength of the high-priority representation. In other words, on trials when the low-priority action plan was cued, but the high-priority action plan was strong, this led to slower responses and more errors.

We note again that when the high-priority item was selected, the strength of all representations correlated with better behavior. In contrast, with the low-priority item, only the strength of the conjunctive representation of the low-priority item was predictive of behavior when that action was cued. There was no evidence of this relationship for the strength of the low-priority response or stimulus representation. Furthermore, the strength of the representations of the low-priority action mostly did not interfere with selection of the other, high-priority action, except for the conjunction on RTs, although a quantitatively similar pattern was observed for the response representation.

Taken together, during selection for the output, the conjunction and response representations, but not stimulus representations, were selectively activated (*Figure 4*) and influenced efficient response selection (*Figure 5*) contingent on the priority and output gating demand of corresponding actions.

**Table 2.** Cluster-level statistics for the main effects of the interaction model between test type and priority regressed on the representational similarity analysis (RSA) scores.

| RSA scores | Effect | Time (ms) | Cluster *T*-value | Cluster p-value |
|---|---|---|---|---|
| Stimulus | Priority | 1566–1598 | 20.11 | <0.01 |
| | Priority | 1846–1866 | 12.45 | <0.01 |
| Conjunction | Test type | 1898–2050 | 98.36 | <0.01 |
| Response | Priority | 1738–1794 | 38.70 | <0.01 |
| | Test type | 1858–1894 | 22.94 | <0.01 |
| | Test type | 2014–2110 | 67.02 | <0.01 |

Note: The clusters were identified from the model that included the effect of priority and test type and their interaction using a cluster-forming threshold, p<0.05. The interaction effect is shown in *Figure 4*.

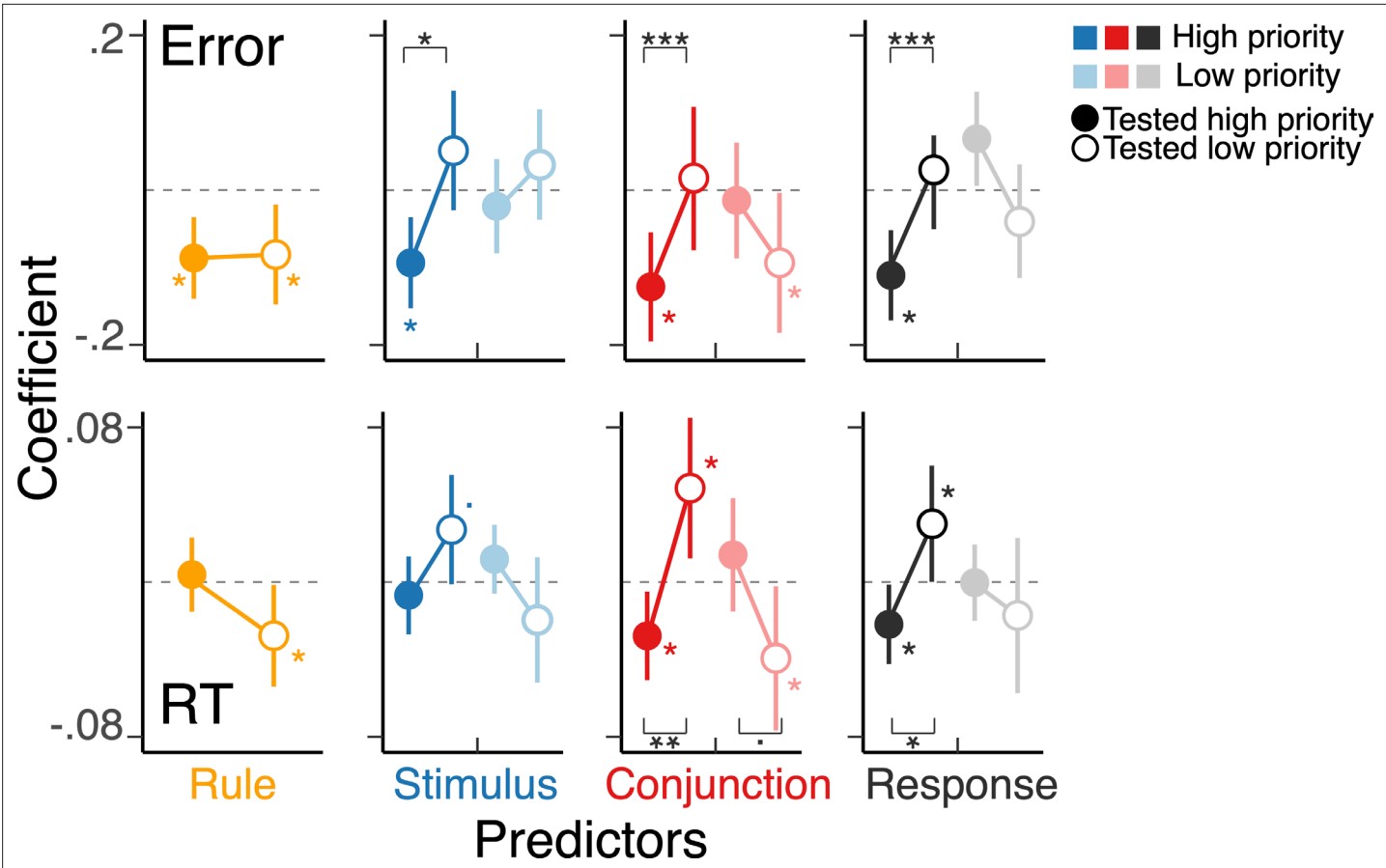

**Figure 5.** Interference between tested and untested actions with different levels of priority. The coefficients of the multilevel regression models predicting the variability in trial-to-trial RTs and errors. The model included representational similarity analysis scores of all action features of both actions during the test phase as predictors as well as the main effect of text context. The left-half side of a panel (denoted as 'H') for stimulus, conjunction, and response corresponds to the features of high-priority action, and the right-half side (denoted as 'L') shows the features of low-priority action. The stars without a bracket indicate the level of significance for individual coefficients, and the stars with a bracket show the effect of selection (i.e. an action required for the test). *, **, ***, and a dot indicate p<0.05, p<0.01, p<.001, and p<0.1, respectively.

These effects were asymmetric such that representations of high-priority action tended to have a stronger impact on output selection, suggesting prioritization is established during output gating.

## Discussion

Our ability to prepare and select from a pool of action plans allows us to rapidly adapt to diverse and changing situations. Theories of cognitive control posit that an integrated control representation that incorporates a conjunction of task-relevant information, such as goals, action rules, sensorimotor features, and rewards, may be critical for flexible action selection (*Hommel, 2019*; *Frings et al., 2020*; *Logan, 1989*). And, indeed, recent evidence has established the critical role of conjunctive representations in execution and control of responses (*Kikumoto and Mayr, 2020*; *Kikumoto et al., 2022*).

If conjunctive representations function as basic building blocks of action preparation and selection one would expect that people can maintain multiple conjunctions in working memory, prioritize them based on their expected utility, and selectively output one or another as circumstances dictate. The current study adapted a previously established paradigm for tracking conjunctive representations in order to test this hypothesis. We found evidence both for simultaneous preparation of two conjunctive representations and for priority-modulated output gating of the conjunction that correspond with the currently targeted action. Furthermore, we found that conjunctive representations influence post-encoding output selection over and above the constituent representations, which were consistently

targeted by attentional selection throughout a trial to maintain, prioritize, and select an action from the candidates held in working memory.

Our results add to a growing body of evidence that maintenance of sensory information for upcoming output selection concurrently transforms task-relevant sensory memories in a more action-oriented or proceduralized representational format (*Oberauer, 2009*; *Hommel, 2019*; *Brass et al., 2017*). Preparing future context-dependent actions recruited conjunctive representations that comprise the unique action constellation for output even after responses are fully specified (*Figures 1, 4 and 5*). Moreover, several current theoretical proposals for working memory suggest that encoded sensory information is reconfigured to a use-optimized format when a memorandum is assigned a specific use or selected for output (such as by retro-cueing) (*Myers et al., 2017*; *Nobre and Stokes, 2019*; *Orhan and Ma, 2019*). Consistent with these hypotheses, we observed here that participants can retrospectively prioritize and select conjunction representations based on task demands.

Importantly, whereas both action conjunctions were maintained throughout the delay, we did not find evidence that the responses were specified until the test period. Though we don't rule out that in certain tasks responses would be maintained without the full conjunction, our results provide evidence that the integrated conjunctive representation is used as a format for holding planned actions in working memory, even when all information is available to specify the response. In this context, we should note that it is possible that the short delay period (750 ms) used in this study may have limited the opportunity for a response representation to be specified during the maintenance period. However, our previous studies that did not include a delay period *Kikumoto and Mayr, 2020*; *Kikumoto and Mayr, 2020* found that response representations were detected less than 300 ms after the response is specified. This suggests that responses can be formed far faster than the maintenance period provided in this study. Nonetheless, decisive dissociation of the timing of representation types could be investigated by temporally varying preparation intervals in future studies.

Notably, unlike prior studies that manipulate post-encoding selection from working memory (*Griffin and Nobre, 2003*; *Souza and Oberauer, 2016*; *Ester et al., 2018*), we observed that stimulus representations showed a relatively weak effect of priority and selection demand and a less of interference on behavior (*Table 2*; *Figure 4*; *Figure 5*). This may be partially driven by the fact that the sensory inputs were explicitly contextualized or transformed by the action rules for the task. However, another mutually compatible account of this discrepancy is that our RSA regression approach allows us to directly test conjunctive representations competing against other constituent formats, such as the simple stimulus representations.

Instead of other action features, our results showed conjunctive representations and response representations reflect the priority of planned actions during post-encoding selection. Consistent with recent theories that explained post-encoding priority effects by linking its informational content to associated actions through attention (*Allport et al., 1987*; *Souza and Oberauer, 2016*; *Olivers and Roelfsema, 2020*; *González-García et al., 2020*), these results suggest that more context-specific representations may be a better format to capture control processes for post-encoding selection of a planned action. Unlike typical working memory tasks in which enhanced sensory representations suffice for successful completion of the task, realistic context-dependent actions require flexible control over input-output mapping, as is manipulated here. Beyond prioritization and selection, optimal control of actions may recruit conjunctive representations that minimally defines relevant task states to encode response representations that execute actions (*Todorov, 2004*; *McNamee and Wolpert, 2019*).

The importance of the integrated representation in our results also fits with recent theories about the geometry of task representations by nonlinear integration of information, wherein sensory and context signals are projected into a high-dimensional population code, as a key computational mechanism for working memory and cognitive flexibility (*Fusi et al., 2016*; *Buschman, 2021*; *Badre et al., 2021*; *Jazayeri and Ostojic, 2021*; *Freund et al., 2021*). Supporting this view, neurons in association cortex (e.g. prefrontal and parietal cortex) exhibit nonlinear mixed selectivity or coding of task feature information in a conjunctive manner. At the population level, this coding aids behavioral flexibility by discriminating readout for diverse combinations of inputs (*Rigotti et al., 2013*; *Parthasarathy et al., 2019*; *Bernardi et al., 2020*; *Panichello and Buschman, 2021*).

A recent model of working memory explicitly addressed conjunctive coding by task-agnostic random networks and highlighted their role in flexibly maintaining arbitrary inputs (*Bouchacourt and Buschman, 2019*). This model proposed that the flexible yet capacity-limited nature of working

memory may originate from the interaction between a structured layer containing subnetworks of sensory features and an unstructured (random) layer responding to integrated information in high-dimensional spaces. Perhaps consistent with these perspectives, the observed conjunctive representations during preparation of future actions may reflect a working memory system that maximizes flexibility by relying on integrative connections (*Buschman, 2021*). For future studies, assessing the adverse effect of working memory load on conjunctive representations could highlight how the capacity limit of working memory constraints integration.

A further benefit of high-dimensional representations is that they can cast multiple task features into separable patterns, which could reduce their overlap and mitigate task interference (*Musslick et al., 2020*; *Badre et al., 2021*). In our task, the rule always overlapped between two action plans, thus action preparation could be aided by heightened separability in high-dimensional representational geometry. Yet, we observed that when prioritized conjunctive representations were strong during the test phase, they interfere with selecting an alternate action (i.e. the low-priority action), perhaps suggesting either that the separability between the two action plans was compromised or not complete. Though, it is also possible that the interference reflects downstream processes that must select between the action plans, as discussed below. Future studies should more directly investigate representational geometries encompassing two action plans and relate to the interference costs, including conditions that do not form integrated representations.

An important question left open by this work is what mechanism enacts prioritization and selection of these conjunctive representations. A context-dependent control mechanism that selectively gates into and out of working memory has been extensively studied as one such mechanism (*O Reilly and Frank, 2006*; *Hazy et al., 2022*; *Frank and Badre, 2012*; *Badre and Frank, 2012*). One computational account, named the PBWM (prefrontal cortex basal ganglia working memory) model, implements gating of working memory via dopaminergic reinforcement learning in basal ganglia nested within cortico-striatal hierarchical loops. The computational architecture of the PBWM model provides a natural mechanism for combining values or expected utility of information into gating decisions. At the algorithmic level, the current study supports the use of selective output gating for action preparation. Yet, unlike previous studies focusing on the role of compositional representations of the task-relevant inputs (e.g. rule or context representations), our results highlighted an active role of integrated representations in biasing prioritization and selection of the planned actions on behavior.

Our results showed that multiple action plans could be retrospectively selected from within working memory, albeit with interference costs from the prioritized action during output selection phase.

We note that within the PBWM framework, the interference effect could be accounted for by changes in expected value that occur after the 'GO!' cue modulating gating operations in the basal ganglia (e.g. changes in drift rate), which may occur regardless of the separability of cortical representations. The late appearance of the priority effect and its interaction with selection demand (*Figure 4*) and more pronounced interference effect of high-priority action features (*Figure 5*) suggest that output selection of a low-priority action may induce switching of output gate that propagates down to the conflict at motor gating. From this perspective, the priority of actions is mainly reflected during gating operations for selecting one or another action. Consistent with this view, most errors were execution of the uncued actions (*Figure 2*).

Furthermore, with the caveat that we do not have statistical evidence of that allows us to draw strong inferences about timing of gating operations over different representations (e.g. conjunctive and response representations) within the test phase, the pattern of the results suggests that the high-priority conjunction is sustained until late, even when it is untested. This sustained pattern is notable in light of our observation that the strength of the high-priority conjunctive representation interferes behavior when the low-priority item is tested, but not vice versa. If verified, it is possible that this coactivation could be the basis of conflict for circuits deciding what to gate, which produces a behavioral interference effect. Though the present study cannot provide direct support for the involvement of these biological gating circuits in the prioritization and selection of conjunctive representations, future studies could explicitly test these hypotheses using measures other than EEG and also could model the computational role of conjunctive representations in gating control of working memory.

Understanding cognitive flexibility requires answering how action plans are represented, maintained, prioritized, and selected based on changing circumstances in our environment. The current study provided an initial evidence that multiple future actions could be maintained as conjunctive

representations, and this level of representation is a key target of processes that prioritize prepared actions and guide their retrospective selection.

## Materials and methods

### Participants

Twenty-six participants (16 females; mean age: 23.2 years) were recruited from the University of Oregon student body. All participants had normal or corrected-to-normal vision, and had no history of neurological or psychiatric disorders. They were compensated $10 per hour plus additional performance-based incentives (see Behavioral procedure section). Participants underwent informed consent using procedures approved by the Human Subjects Committee at the University of Oregon. After preprocessing the EEG data, two participants were removed and were not analyzed further due to excessive amounts of artifacts (i.e. more than 25% of trials; see EEG recordings and preprocessing for detail).

### Behavioral procedure

For each trial, participants were instructed to prepare two possible actions for upcoming tests by applying the single shared rule to two independent stimuli (*Figure 1a*). The required two actions on each trial were selected from the shared action constellations or S-R mappings shown in *Figure 1b*.

At the beginning of the *encoding* period (0–750 ms; *Figure 1a*), four crosses appeared at each of the four corners of a white square frame (6.6° in one side). Two of these locations were marked as to be remembered by coloring one of their cross bars either blue or green for 200 ms, as shown in *Figure 1a*. The different colors of bars cued the priority of each action in terms of its probability of being the target at the upcoming test and so were task-relevant features. Each bar was randomly assigned to be in a vertical or horizontal orientation, which allowed us to randomize and orthogonalize the locations of stimuli by including trials with an overlap of locations in 25% of trials. Then, following 550-ms retention interval with only a fixation cross, one of the three possible action rules was randomly cued for 200 ms followed by a 550-ms delay. This marked the *preparation* period (750–1500 ms; *Figure 1a*) during which participants could prepare both actions based on combination of the rule and remembered locations. Each action rule ('vertical', 'horizontal', and 'diagonal') uniquely mapped the four stimulus positions to four response keys that were arranged in 2×2 matrix (4, 5, 1, and 2 on the number pad; *Figure 1b*). For example, the horizontal rule mapped the left-top bar to the right-top response and the right-bottom bar to the left-bottom response as the correct set of responses. To focus on the effect of simultaneously planning two independent actions, we excluded trials during which two actions led to the same response due to the overlap of stimulus positions during analyses. We ruled out any concern regarding the interference effect based on the compatibility of the target stimulus and the action rule having matching features (see details for the Supplementary information).

In order to encourage participants to prepare two actions before the final *test* period (starting 1500 ms after the stimulus, *Figure 1a*), an adaptive cutoff was used to limit the test interval. The test screen showed a cue 'GO!' in either blue or green to prompt participants to execute one of the prepared actions of the corresponding color. Responses were allowed to register only within a limited cutoff time interval following the cue. This cutoff interval was initially defined as 1200 ms, then it was adaptively adjusted trial-to-trial by increasing the interval after any incorrect trials whereas decreasing the interval after five consecutive correct trials. The step size of adjustment of the interval was randomly selected from 11.8 ms, 23.5 ms, or 35.3ms.

In order to manipulate priority, participants were explicitly instructed to prepare and maintain two actions simultaneously but to expect different probabilities of each action being tested. For example, in the example block depicted in *Figure 1a*, the action associated with the green color bar would be tested on 70% of trials, whereas the action associated with the blue color would be tested on 30% of trials. Thus, green cued the high-priority target, and blue cued the low-priority target. Expected test probability was cued by color of the bar stimuli (i.e. blue and green) to either a high (70%) or low (30%) test probability. The assignment of color to test probability was randomized for every block, therefore, one action plan was always more likely to be tested than the other in all trials. To prevent participants from disregarding information related to the low-test probability action, participants gained performance-based incentives only when the overall accuracy was above 85% in a given

block, which was difficult to achieve if they used such a strategy, given the speeding required by the adaptive cutoff.

There were two practice blocks and 180 experimental blocks. Each block lasted 25 s, during which participants were instructed to complete as many trials as possible. Trials that were initiated within the 25-s block duration but extended beyond it were allowed to finish. Throughout the experimental session, participants were reminded to respond as accurately and fast as possible within adaptive cutoff intervals. Feedback about overall accuracy and the amount of performance-based incentives accrued was provided at the end of each block. All stimuli were generated in Matlab (Mathworks) using the Psychophysics Toolbox (*Brainard, 1997*) and were presented on a 17-inch CRT monitor (refresh rate: 60 Hz) at a viewing distance of 100 cm.

### EEG recordings and preprocessing

We recorded scalp EEG activities using 20 tin electrodes on an elastic cap (Electro-Caps) using the international 10/20 system. The 10/20 sites F3, Fz, F4, T3, C3, Cz, C4, T4, P3, PZ, P4, T5, T6, O1, and O2 were used along with five nonstandard sites: OL halfway between T5 and O1; OR halfway between T6 and O2; PO3 halfway between P3 and OL; PO4 halfway between P4 and OR; and POz halfway between PO3 and PO4. Electrodes placed ~1 cm to the left and right of the external canthi of each eye recorded horizontal electrooculogram (EOG) to measure horizontal saccades. To detect blinks, vertical EOG was recorded from an electrode placed beneath the left eye. The left mastoid was used as reference for all recording sites, and data were re-referenced off-line to the average of all scalp electrodes.

The scalp EEG and EOG were amplified with an SA Instrumentation amplifier with a bandpass of 0.01–80 Hz, and signals were digitized at 250 Hz in LabView 6.1 running on a PC. EEG data was first segmented into 27.5-s intervals to include all trials within a block. After time-frequency decomposition was performed, these epochs were further segmented into smaller epochs for each trial using the time interval of –200–2200 ms relative to the onset of stimuli (*Figure 1a*). The trial-to-trial epochs including blinks (>250 $\mu v$, window size = 200 ms, window step = 50 ms), large eye movements (>1°, window size = 200 ms, window step = 10 ms), blocking of signals (range = –0.01–0.01 $\mu v$, window size = 200 ms) were excluded from subsequent analyses. Using these criteria, we excluded 268 trials (15%) of trials in each subject on average. We retained an average of 1100 trials for the condition where the high-priority action was tested, and 472 trials for the low-priority action test condition (60 or 25% of the original data, respectively).

### Time-frequency analysis

Temporal-spectral profiles of single-trial EEG data were computed via complex wavelet analysis (*Cohen, 2014*) by applying time-frequency analysis to preprocessed EEG data epoched for the entire block (>27.5 s to exclude the edge artifacts). The power spectrum was convolved with a series of complex Morlet wavelets, where $t$ is the time, $f$ is the frequency increased from 1 to 35 $Hz$ in 35 logarithmically spaced steps, and σ defines the width of each frequency band, set according to $n/2pft$, where $n$ increased from 3 to 10. We used logarithmic scaling to keep the width across frequency band approximately equal, and the incremental number of wavelet cycles was used to balance temporal and frequency precision as a function of frequency of the wavelet. After convolution was performed in the frequency domain, we took an inverse of the Fourier transform, resulting in complex signals in the time domain. A frequency band-specific estimate at each time point was defined as the squared magnitude of the convolved signal for instantaneous power.

### Representational similarity analysis

We decoded action-relevant information in a time-resolved manner following our previously reported method with a few modifications (*Kikumoto and Mayr, 2020*). As the first step, separate linear decoders were trained to classify all possible action constellations (*Figure 1b*) for each independently planned action at every sample in trial-to-trial epochs. Specifically, we performed a penalized linear discriminant analysis using the caret package in R (*Hastie et al., 1995*; *Kuhn, 2008*). For each action, the 12 unique action constellations were defined by the combination of 3 rules and 4 stimulus positions (*Figure 1b*). This step produced a graded profile of classification probabilities (i.e. a confusion

matrix) for each action constellation, reflecting the multivariate distance of neural patterns between action plans.

Decoders were trained with the instantaneous power of rhythmic EEG activity, which was averaged within the predefined ranges of frequency values (1–3 Hz for the delta-band, 4–7 Hz for the theta-band, 8–12 Hz for the alpha-band, 13–30 Hz for the beta-band, 31–35 Hz for the gamma-band), generating 100 data features (5 frequency-bands × 20 electrodes) for training. Within individuals and frequency bands, the data points were z-transformed across electrodes to remove effects that uniformly influenced all electrodes. We used a k-fold repeated, cross-validation procedure to evaluate the decoding results by randomly partitioning single-trial EEG data into four independent folds with equal number of observations of each action constellation. All trials including error trials were used as the training sets. After all folds served as the test sets, each cross-validation cycle was repeated eight times, in which each step generated a new set of randomized folds. Resulting classification probabilities were averaged across all cross-validated results with the best-tuned hyperparameter to regularize the coefficients for the penalized linear discriminant analysis.

We next performed RSA on the graded classification probabilities to assess the underlying similarity structure. Each RSA model matrix uniquely represents a potential, underlying representation (e.g. rules, stimuli, responses, and conjunctions), which makes unique predictions for different action plans. Specifically, for independent decoding results of each of the action plans, we regressed the vector of logit-transformed classification probabilities onto RSA model vectors. To estimate the unique variance explained by competing models, we regressed all model vectors simultaneously, resulting in coefficients for each of the four model vectors. We also included subject-specific regressors of z-scored, average RTs, and accuracies in each action constellation to reduce potential biases in decoding of conjunctions. These coefficients, expressed in their corresponding t-values, reflect the quality of action representations at the level of single trials, which was later related to variability in behavior. We excluded t-values that exceeded five SDs from means for each sample point, which excluded less than 1% of the entire samples. The resulting t-values were averaged over 40 ms non-overlapping time windows for visualization (*Figure 3*).

## Multilevel modeling and cluster-based permutation test

We used multilevel models to further assess the decoded representations (*Figure 4*; *Table 1*) and relate them to trial-to-trial variability in behavior (*Figure 5*). For these analyses, we used time-resolved and time-averaged RSA scores of both the basic action features and the conjunctions. For the results of time-resolved regression (*Figure 4*), we used non-parametric permutation tests to evaluate the decoding results in the time domain (*Maris and Oostenveld, 2007*). First, we performed a series of regression analysis over time and identified samples that exceeded threshold for cluster identification (cluster-forming threshold, p<0.05). Then, empirical cluster-level statistics were obtained by taking the sum of t-values in each identified cluster with consecutive time points. Finally, nonparametric statistical tests were performed by computing a cluster-level p-value (cluster-significance threshold, p<0.05, two-tailed) from the distributions of cluster-level statistics, which were obtained by Monte Carlo iterations of regression analysis with shuffled condition labels. When we regressed trial-to-trial response times and accuracies on RSA scores of action features, the coefficients were averaged across three time windows (as described in Behavioral procedure) that were selected a priori: the encoding period, the preparation period, and the test period. These periods occurred 0–750 ms, 750–1500 ms, and 1500–2200 ms relative to the onset to the stimuli, respectively, were used as predictors in the regression model. For all models, subject-specific intercept and slopes were included as random effects. We report the test statistics, coefficients (beta), and confidence intervals as a metric of statistical reliability because the appropriate way to calculate degrees of freedom in multilevel models is still being debated. For models predicting accuracies, we used multilevel logistic regression. For models predicting response times, error trials were excluded.

## Acknowledgements

We would like to acknowledge the Cognitive Dynamics lab, the Badre Lab and the Human Cognition and Learning lab, particularly Jiafan Jia, Apoorva Bhandari and Haley Keglovits for helpful comments and discussion. This project was supported by funding from the National Institute of Mental Health (R01 MH125497), the National Institute of Neurological Disorders and Stroke (R21

NS108380), and a Multidisciplinary University Research Initiative award from the Office of Naval Research (N00014-16-1-2832) to DB and from the National Science Foundation Grant(1734264) to UM.

## Additional information

### Competing interests

David Badre: Reviewing editor, *eLife*. The other authors declare that no competing interests exist.

### Funding

| Funder | Grant reference number | Author |
|--------|------------------------|--------|
| National Science Foundation | 1734264 | Ulrich Mayr |
| National Institute of Mental Health | MH125497 | David Badre |
| National Institute of Neurological Disorders and Stroke | NS108380 | David Badre |
| Multidisciplinary University Research Initiative | N00014-16-1-2832 | David Badre |

The funders had no role in study design, data collection and interpretation, or the decision to submit the work for publication.

### Author contributions

Atsushi Kikumoto, Conceptualization, Data curation, Formal analysis, Writing - original draft, Writing - review and editing; Ulrich Mayr, Conceptualization, Supervision, Funding acquisition, Writing - original draft; David Badre, Conceptualization, Supervision, Funding acquisition, Writing - original draft, Writing - review and editing

### Author ORCIDs

Atsushi Kikumoto ⓘ http://orcid.org/0000-0002-2179-2700
Ulrich Mayr ⓘ http://orcid.org/0000-0002-7512-4556
David Badre ⓘ http://orcid.org/0000-0001-7538-6241

### Decision letter and Author response

Decision letter https://doi.org/10.7554/eLife.80153.sa1
Author response https://doi.org/10.7554/eLife.80153.sa2

## Additional files

### Supplementary files

- MDAR checklist

### Data availability

Data and analyses are available through OSF (https://osf.io/4mx8c/). Specifically, the repository contains trial-by-trial behavioral data files, and all relevant EEG data.

The following dataset was generated:

| Author(s) | Year | Dataset title | Dataset URL | Database and Identifier |
|-----------|------|---------------|-------------|-------------------------|
| Kikumoto A, Mayr U, Badre D | 2022 | The Role of Conjunctive Representations in Prioritizing and Selecting Planned Actions | https://osf.io/4mx8c/ | Open Science Framework, 4mx8c |

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

# Appendix 1

## Supplementary information

### Compatibility effect between rules and stimuli by shared features

The two of the action rules (i.e. 'horizontal' vs. 'vertical') and stimuli (i.e. horizontal and vertical bars) shared the feature in an abstract way in some trials. Such an overlap could have caused an interference such as an involuntary shift of attention to the stimulus where the colored bar had the matching orientation, which could influence behavioral and decoding results. To rule out these possibilities, we assessed the effect of compatibility of the tested item and the cued rule, excluding trials with a diagonal rule.

On behavior, we found no evidence of an effect of compatibility (mean RT: 383 ms [4.52] for compatible and 383 ms [4.52] for incompatible, $t(1,23)$ = .27, beta = .47, 95% CI [–.022.029]; mean error rate: 13.8% [.97] for compatible and 14.2% [.97] for incompatible, $t(1,23)$ = –.14, beta = .006, 95% CI [–.091.079]; mean adaptive cutoff: 862 ms [7.38] for compatible and 860 ms [7.38] for incompatible, $t(1,23)$ = –.52, beta = .006, 95% CI [–.091.079]; mean swapping error rate: 75.1% [4.74] for compatible and 75.9% [4.74] for incompatible, $t(1,23)$ = .62, beta = .079, 95% CI [–.170.328]).

As the mapping between the priority of actions and the bar orientations was randomized across trials, the RSA-based decoding of actions defined by rules and stimulus positions should be insensitive even if there was an effect of compatibility. Indeed, we found no effect of compatibility on the decoding of conjunctions during both maintenance period (high-priority conjunction: $t(1,23)$ = –.71, beta = .02, 95% CI [–.043.020]; high-priority conjunction: $t(1,23)$ = –.52, beta = .005, 95% CI [–.040.023]) and test period (high-priority conjunction: $t(1,23)$ = –.99, beta = .01, 95% CI [–.048.015]; high-priority conjunction: $t(1,23)$ = –.80, beta = .008, 95% CI [–.018.045]). Furthermore, we found that the decoding of the bar orientation was at chance level during the interval when we observe evidence of the conjunctive representations. Thus, we conclude that the compatibility of the stimuli and the rule did not contribute to the decoding of conjunctive representations and behavior.

### Output gating of action representations during the test phase: response-aligned

The results presented in the main text are based on EEG data that are aligned to the onset of stimuli (*Figure 1*). To closely investigate the timing of modulation of stimulus, conjunction, and response representations around the timing of response execution, we performed the RSA-derived decoding analysis by using response-aligned EEG data. Because of relatively short RTs observed in the current study (due to the adaptive deadline procedure and the long preparation period), it is important to characterize the temporal dynamics of modulation effect on relevant representations relative to a moment of response selection. Consistent to the stimulus-aligned result (*Figure 4*), we found that both conjunctive representations and response representations were modulated by the priority of actions and the test type. Notably, for the conjunctive representations in the low-priority-tested condition, the effect of dropping high-priority conjunction emerges late compared to the enhancement of low-priority conjunction. This timing of reversal has contributed to the late timing of the interaction effect seen in the stimulus-aligned result.

### How distributed and frequency-specific are RSA results?

We examined how different frequency bands contribute to the decoding of both conjunctions and constituent features as an exploratory analysis. We performed search-light RSA analyses to subsample training data in the frequency and channel space. Specifically, to investigate the frequency space, we replicated the combination of decoding and RSA analyses separately for the delta (1–3 Hz), theta (4–7 Hz), alpha (8–12 Hz), beta(13–30 Hz), and gamma (31–35 Hz) frequency bands. For the channel (electrodes) space, we defined ROIs as a group of electrodes of F3-C3-T3, F4-C4-T4, Fz-Cz, OL_O1, OR-O2, P3-PO3-T5, P4-O4-T6, and Pz-POz. To reduce the influence of temporal smearing, which could differ across frequency bands, we averaged data over a priori selected time intervals (i.e. encoding, preparation, and test phase) prior to the training of decoders. Note that by averaging signals, we focus on the informative EEG patterns that are temporally stationary. Other steps in the analysis were identical to the one described in Representational similarity analysis section.

Across the encoding, maintenance, and selection phases, participants exhibited considerable variability in terms of the frequency bands in which specific information was expressed. We did not find any systematic differences among individuals in the channels differentially contributing to code

action representations (not shown). However, in the frequency space, the following points seem to be consistent across individuals: (1) stimulus positions and rules (i.e. cue words) were coded in the theta and alpha-band when they are visually presented and (2) conjunctions tended to be expressed most strongly in the delta band. These results are overall consistent with the previous study (*Kikumoto and Mayr, 2020*).

