## [Editor Report]

This study presents important findings regarding the control processes that underlie planned actions. The evidence for the authors' conclusions is compelling, using state-of-the-art model-based decoding analysis. This work is of interest to cognitive scientists spanning the subfields of cognitive control, working memory, and motor control.

---

## [Decision Letter]

**Decision letter after peer review:**

Thank you for submitting your article "The Role of Conjunctive Representations in Prioritizing and Selecting Planned Actions" for consideration by *eLife*. Your article has been reviewed by 3 peer reviewers, and the evaluation has been overseen by a Reviewing Editor and Floris de Lange as the Senior Editor. The following individual involved in the review of your submission has agreed to reveal their identity: Daniel Schneider (Reviewer #2).

Essential revisions:

1) Reviewer 1 raised the possibility of a potential compatibility effect that might be influencing the results reported. These stimulus-rule configurations need to be investigated and discussed in detail.

2) There were concerns about the results reported in Figure 4 and their interpretation that need to be addressed. Specifically, there was some doubt as to whether these results reflect output gating and there were some methodological inconsistencies.

3) More detail is needed regarding the methods used. The reviewers appreciated that many of the methods were reported previously, but many of these details need to be included here to allow the reader to fully understand the approach.

*Reviewer #1 (Recommendations for the authors):*

Methods: Behavioral procedure section

If I'm not mistaken, the fact that either the horizontal or vertical bar of the cross-shaped stimuli was colored is irrelevant to the task. I wonder whether stimuli consisting of bars with distinct orientations were an ideal choice because horizontal and vertical bars together with two of the action rules (i.e., "horizontal" vs. "vertical") might have created unwanted interactions in most trials. Two types of interactions seem possible: at cue time horizontal and vertical cues might have caused an involuntary shift of attention to the stimulus where the colored bar had the matching orientation (in Figure 1A the cue "horizontal" might cause attention to be directed to the bottom-right stimulus where the blue horizontal bar pops out). At the time of the go signal it is possible that Stroop-like interferences occurred (e.g., in Figure 1A the green "GO" signal requires that participants push a button that is horizontally adjacent to the top-left stimulus although the stimulus contains an incongruent green vertical bar). It is possible that attentional effects (validly vs. invalidly cued stimuli) and Stroop-like effects (congruent vs. incongruent stimuli) were reflected in reaction times and in the speed at which representations arose. Given such potentially systematic differences, it would be good to know whether they contributed to the explained variability of any of the RSA representations – or preferably not. Regressing out RSAs based on reaction times and accuracies might have helped to reduce the risk but I'm not sure whether this filtered out potential confounds entirely.

Methods: Representational Similarity Analysis

Please mention which kind of decoders were used.

Results: Action Representations during Early Encoding and Preparation Phase, 2nd paragraph

I'd suggest reporting p-values and effect sizes. Initially, I mistook the b-values for Bayes factors. The statistics for conjunction representations during preparation seem to be rather weak. Also, why not report the statistics for the test phase? The next section talks about the test phase but focuses on the results in Figure 4.

Results: Action Representations during Early Encoding and Preparation Phase, 3rd paragraph

The authors found no significant effect of test probability on representations. For factors stimulus and rule that seems to reflect that there was no prioritization. However, for conjunctions, it is difficult to draw such a conclusion given the poor signal-to-noise ratio.

Results: Output gating of Action Representations During the Test Phase, 2nd paragraph

The authors state: "Specifically, high priority items were more active than low priority items when they were cued, but the reverse was the case when low priority items were cued."

I'm having difficulties seeing the mentioned reverse effect. For stimulus representations it seems to not occur. For conjunction and response representations the reverse effect is somewhat transient. Also, in Table 1 it seems surprising that the t-value for stimulus (low priority) /low priority tested is as large as 2.19 while in Figure 5 the corresponding curve undulates around zero. Perhaps I misunderstood how Figure 5 and Table 1 relates to one another. Please clarify.

I'm not sure whether the absence of correlations between high- and low-priority conjunctions is necessarily indicating independent representations that don't compete with one another. Also, wouldn't independent representation contradict what is said in the previous paragraph that high-priority plans interfere with low-priority plans? Furthermore, the interaction for the conjunction representation arises later than the representation for the response. This seems to be non-intuitive given that the response, in the end, is what is behaviorally required. Please elaborate.

Results: Figure 5 and corresponding text

Please explain why "facilitatory effects were diminished or reversed into interference when the low priority item was tested as a function of the strength of the high priority representation." From what I gather there is a more negative correlation between reaction times and errors for low-priority representations when they are actually tested, or more generally speaking: there seems to be a more negative correlation for low- as well as for high-priority representations when these representations are tested. So then, I don't understand why the correlations for high- and low-priority representations reflect an asymmetry or interference. Also, it is stated that there is no correlation for low-priority representations but, once again, I'm not sure whether any conclusions can be derived here given the poor signal-to-noise ratio. In addition, the representations in the left vs. right column should only differ after 1500ms. I see quite a bit of difference that might be important to account for in any statistical analysis.

*Reviewer #2 (Recommendations for the authors):*

In essence, the authors should be more careful in interpreting their results. While I think that a clever experimental design was used, I also think that quite a lot of processing steps are necessary to reach the given conclusions. Therefore, the authors need to pay more attention to justifying all of these processing steps (e.g., see different p-thresholds).

Furthermore, I find the results regarding the interaction of test-type and priority factors unconvincing in their current form. The time course of these effects does not seem very plausible to me. Here, the authors should highlight inconsistencies more clearly in the discussion of the results.

It is very important to emphasize more clearly on what basis the authors determined the individual RSA model vectors (stimulus, rule, conjunction, response) and how independence could be guaranteed here. This is mentioned briefly in the methods section. For readers who are not particularly familiar with the application of RSA on the basis of decoding result patterns (and I would honestly count myself among them), this part is unfortunately not particularly easy to understand. More background information on the combination of these methods would be necessary here (especially regarding the RSA model vectors and the multilevel models).

More details are required in the Methods section:

What were the performance-based incentives about?

The adaptive response interval was decreased after each correct trial and additionally decreased after five correct trials? Or was it increased after incorrect trials and decreased after five correct trials? The respective sentence seems quite confusing to me.

How many epochs were excluded from the analysis? What was the number of epochs remaining for each experimental condition?

Reviewer #3 (Recommendations for the authors):

1. "Conjunctive representations were significantly modulated by the selection demand (i.e., cued as an output or not) dependent on their priority (Figure 4; see Table 2 for the main effects and Supplementary Figure 1 for the results using response-aligned data)". However, when looking at Figure 4, it seems that in the "tested low priority" condition, the conjunction representation strength of low priority action becomes higher than high priority action (middle row) only after the representation of the low priority response passes the high priority response (bottom row). The same pattern can also be observed in Figure 4 – supplement 1, in which t-values are time-locked to responses. If the conjunction is used to produce the response, its "cross-over" should precede the "cross-over" of responses. Given this analysis is key to the authors' conclusion that the conjunction representation is subject to output gating, the authors need to address this order issue by looking into a formal statistical test of the order of the two cross-over effects (or the order of the beginning of the two priority x test interactions) and further discussing the functional importance of the cross-over of the conjunction (e.g., whether the change is related to performance in the current trial or post-performance monitoring/adjustment).

2. Is it possible to explore the key channels and/or frequency bands that are important in decoding the conjunction representations?

---

## [Author Response]

Essential revisions:1) Reviewer 1 raised the possibility of a potential compatibility effect that might be influencing the results reported. These stimulus-rule configurations need to be investigated and discussed in detail.

We agree that it is important to rule out any effects of involuntary attention that might have been elicited by our stimulus choices. To address the Reviewer’s concern, we conducted control analyses to test if there was any influence of Stroop-like interference on our measures of behavior or the conjunctive representation. Specifically, we assessed the effect of compatibility of the tested item (i.e., “horizontal” and “vertical” bar) and the cued rule, excluding trials with a diagonal rule.

We found no evidence of an effect of compatibility on behavior (mean RT: 383ms (4.52) for compatible and 383ms (4.52) for incompatible, *t*(1,23) = .27, *β* = .47, 95% CI [-.022.029]; mean error rate: 13.8% (.97) for compatible and 14.2% (.97) for incompatible, *t*(1,23) = -.14, *β* = .006, 95% CI [-.091.079]; mean adaptive cutoff: 862ms (7.38) for compatible and 860ms (7.38) for incompatible, *t*(1,23) = -.52, *β* = .006, 95% CI [-.091.079]; mean swapping error rate: 75.1% (4.74) for compatible and 75.9% (4.74) for incompatible, *t*(1,23) = .62, *β* = .079, 95% CI [-.170.328]).

In addition, we tested whether there is a compatibility effect of viable vs. non-viable direction matching: vertical and horizontal rule vs. diagonal rule. We found no overall difference between the diagonal rule and the other two rules on in the effect of priority of test type on RT, *t*(1,23) = -1.21, *β* = -1.62, 95% CI [-.031.007], cutoff time, *t*(1,23) = 1.21, *β* = 2.70, 95% CI [-.006.026], and swapping error, cutoff time, *t*(1,23) = 1.21, *β* = 2.70, 95% CI [-.006.026], except for error rate, *t*(1,23) = -2.07, *β* = 2.70, 95% CI [-.106 -.003] where the effect of high-test probability decreasing errors was less for trials with the diagonal rule.

As the mapping between the priority of actions and the bar orientations was randomized across trials, the RSA-based decoding of actions defined by rules and stimulus positions should be insensitive even if there was an effect of compatibility. Indeed, we found no effect of compatibility on the decoding of conjunctions during both maintenance period (high priority conjunction: *t*(1,23) = -.71, *β* = .02, 95% CI [-.043.020]; low(?) priority conjunction: *t*(1,23) = -.52, *β* = .005, 95% CI [-.040.023]) and test period (high priority conjunction: *t*(1,23) = -.99, *β* = .01, 95% CI [-.048.015]; low(?) priority conjunction: *t*(1,23) = -.80, *β* = .008, 95% CI [-.018.045]). Furthermore, we found that the decoding of the bar orientation was at chance level during the interval when we observe evidence of the conjunctive representations. Thus, we conclude that the compatibility of the stimuli and the rule did not contribute to the decoding of conjunctive representations or to behavior.

We thank the reviewer for encouraging us to conduct these control analyses, as it has strengthened the results. In the revised manuscript, we note these control analyses in the Materials and methods (pg. 7) and provide details in the supplemental materials.

2) There were concerns about the results reported in Figure 4 and their interpretation that need to be addressed. Specifically, there was some doubt as to whether these results reflect output gating and there were some methodological inconsistencies.

Thank you for raising this important point. First, we fixed reported methodological inconsistencies such as the cluster (*P-*value and *cluster*-forming threshold). Further, we fully agree that the difference in the time course for the response and conjunctive representations in the low priority, tested condition is unexpected and would complicate the perspective that the conjunctive representation contributes to efficient response selection. However, additional analysis indicates that this apparent pattern in the stimulus locked result is misleading and there is a more parsimonious explanation. First, we wish to caution that the data are relatively noisy and likely are influenced by different frequency bands for different features. Thus, fine-grained temporal differences should be interpreted with caution in the absence of positive statistical evidence of an interaction over time. Indeed, though Figure 4 in the original submission shows a quantitative difference in timing of the interaction effect (priority by test type) across conjunctive representation and response representation, the direct test of this four way interaction [priority x test type x representation type (conjunction vs. response), x time interval (1500 ms to 1850 ms vs. 1850 to 2100 ms)] is not significant, (*t*(1,23) = 1.65, *β* = .058, 95% CI [-.012.015]). The same analysis using response-aligned data is also not significant, (*t*(1,23) = -1.24, *β* = -.046, 95% CI [-.128.028]). These observations were not dependent on the choice of time interval, as other time intervals were also not significant. Therefore, we do not have strong evidence that this is a true timing difference between these conditions and believe this is likely driven by noise.

Further, we believe the apparent late emergence of difference in two conjunctions when the low priority action is tested is more likely due to a slow decline in the strength of the untested high priority conjunction rather than a late emergence of the low priority conjunction. This pattern is clearer when the traces are aligned to the response. The tested low priority conjunction emerges early and is sustained when it is the tested action and declines when it is untested (-226 ms to 86 ms relative to the response onset, cluster-forming threshold, *p* <.05). These changes eventually resulted in a significant difference in strength between the tested versus untested low priority conjunctions just prior to the commission of the response (Figure 1, the panel on right column of the middle row, the black bars at the top of panel). Importantly, the high priority conjunction also remains active in its untested condition and declines later than the untested low priority conjunction does. Indeed, the untested high priority conjunction does not decline significantly relative to trials when it is tested until after the response is emitted (Figure 1, the panel on right column of the middle row, the red bars at the top of panel). This results in a late emerging interaction effect of the priority and test type, but this is not due to a late emerging low priority conjunctive representation.

In summary, we do not have statistical evidence of a time by effect interaction that allows us to draw strong inferences about timing. Nonetheless, even the patterns we observe are inconsistent with a late emerging low priority conjunctive representation. And if anything, they support a late decline in the untested high priority conjunctive representation. This pattern of the result of the high priority conjunction being sustained until late, even when it is untested, is also notable in light of our observation that the strength of the high priority conjunctive representation interferes behavior when the low priority item is tested, but not vice versa. We now address this point about the timing directly in the results (pg. 15-16) and the discussion (pg. 21), and we include the response locked results in the main text along with the stimulus locked result including exploratory analyses reported here.

3) More detail is needed regarding the methods used. The reviewers appreciated that many of the methods were reported previously, but many of these details need to be included here to allow the reader to fully understand the approach.

Thank you for the suggestion, we’ve now added more details of the task procedure (pg. 6 and 7) and the decoding method, as well as the motivatatio for including the RSA model of conjunctive representation (pg. 4 and 5).

Reviewer #1 (Recommendations for the authors):Methods: Behavioral procedure sectionIf I'm not mistaken, the fact that either the horizontal or vertical bar of the cross-shaped stimuli was colored is irrelevant to the task.

The color was not irrelevant to the task. But, we now see that these task details were confusing in the original submission. Thus, we will clarify the rationale and taskrelevance of the different bar orientations and color cues both here and in the main text.

On each trial, there was one horizontal bar and one vertical bar each colored differently (red or green). The orthogonal bar orientations allowed us to fully randomize the location of the two stimuli that cued the 2 actions by including trials on which both bars appear at the same location (25% of trials). The different colors of bars were taskrelevant in that they cued its probability of being the target at the upcoming test and so could be used for prioritization. For example, in the example block depicted in Figure 1a, the action associated with the green color bar would be tested on 70% of trials, whereas the action associated with the blue color would be tested on 30% of trials. Thus, green cued the high-priority target and blue cued the low-priority target. We have revised the methods to make these details clearer (pg. 6-7)**.**

I wonder whether stimuli consisting of bars with distinct orientations were an ideal choice because horizontal and vertical bars together with two of the action rules (i.e., "horizontal" vs. "vertical") might have created unwanted interactions in most trials. Two types of interactions seem possible: at cue time horizontal and vertical cues might have caused an involuntary shift of attention to the stimulus where the colored bar had the matching orientation (in Figure 1A the cue "horizontal" might cause attention to be directed to the bottom-right stimulus where the blue horizontal bar pops out). At the time of the go signal it is possible that Stroop-like interferences occurred (e.g., in Figure 1A the green "GO" signal requires that participants push a button that is horizontally adjacent to the top-left stimulus although the stimulus contains an incongruent green vertical bar).

We agree that it is important to rule out any effects of involuntary attention that might have been elicited by our stimulus choices. To address the Reviewer’s concern, we conducted control analyses to test if there was any influence of Stroop-like interference on our measures of behavior or the conjunctive representation. Specifically, we assessed the effect of compatibility of the tested item (i.e., “horizontal” and “vertical” bar) and the cued rule, excluding trials with a diagonal rule.

We found no evidence of an effect of compatibility on behavior (mean RT: 383ms (4.52) for compatible and 383ms (4.52) for incompatible, *t*(1,23) = .27, *β* = .47, 95% CI [-.022.029] ; mean error rate: 13.8% (.97) for compatible and 14.2% (.97) for incompatible, *t*(1,23) = -.14, *β* = .006, 95% CI [-.091.079]; mean adaptive cutoff: 862ms (7.38) for compatible and 860ms (7.38) for incompatible, *t*(1,23) = -.52, *β* = .006, 95% CI [-.091.079]; mean swapping error rate: 75.1% (4.74) for compatible and 75.9% (4.74) for incompatible, *t*(1,23) = .62, *β* = .079, 95% CI [-.170.328]).

In addition, we tested if there is a compatibility effect of viable vs. non-viable direction matching: vertical and horizontal rule vs. diagonal rule. We found that no overall difference between diagonal and the other two rules on in the effect of priority of test type on RT, *t*(1,23) = -1.21, *β* = -1.62, 95% CI [-.031.007], cutoff time, *t*(1,23) = 1.21, *β* = 2.70, 95% CI [-.006.026], and swapping error, cutoff time, *t*(1,23) = 1.21, *β* = 2.70, 95% CI [-.006.026], except for error rate, *t*(1,23) = -2.07, *β* = 2.70, 95% CI [-.106 -.003] where the effect of high-test probability decreasing errors was less for trials with the diagonal rule.

As the mapping between the priority of actions and the bar orientations was randomized across trials, the RSA-based decoding of actions defined by rules and stimulus positions should be insensitive even if there was an effect of compatibility. Indeed, we found no effect of compatibility on the decoding of conjunctions during both maintenance period (high priority conjunction: *t*(1,23) = -.71, *β* = .02, 95% CI [-.043.020]; low(?) priority conjunction: *t*(1,23) = -.52, *β* = .005, 95% CI [-.040.023]) and test period (high priority conjunction: *t*(1,23) = -.99, *β* = .01, 95% CI [-.048.015]; low(?) priority conjunction: *t*(1,23) = -.80, *β* = .008, 95% CI [-.018.045]). Furthermore, we found that the decoding of the bar orientation was at chance level during the interval when we observe evidence of the conjunctive representations. Thus, we conclude that the compatibility of the stimuli and the rule did not contribute to the decoding of conjunctive representations or to behavior.

We thank the reviewer for encouraging us to conduct these control analyses, as it has strengthened the results. In the revised manuscript, we note these control analyses in the Materials and methods (pg. 7) and provide details in the supplemental materials.

It is possible that attentional effects (validly vs. invalidly cued stimuli) and Stroop-like effects (congruent vs. incongruent stimuli) were reflected in reaction times and in the speed at which representations arose. Given such potentially systematic differences, it would be good to know whether they contributed to the explained variability of any of the RSA representations – or preferably not. Regressing out RSAs based on reaction times and accuracies might have helped to reduce the risk but I'm not sure whether this filtered out potential confounds entirely.Methods: Representational Similarity AnalysisPlease mention which kind of decoders were used.

We thank the reviewer for noting this oversight. We have now provided this information in the methods (pg. 9).

Results: Action Representations during Early Encoding and Preparation Phase, 2nd paragraphI'd suggest reporting p-values and effect sizes. Initially, I mistook the b-values for Bayes factors. The statistics for conjunction representations during preparation seem to be rather weak. Also, why not report the statistics for the test phase? The next section talks about the test phase but focuses on the results in Figure 4.

We agree that our use of b-values was confusing in the original submission. To clarify, *b* denoted coefficients of the fixed effects of the multilevel models. To avoid confusion, we replaced *b* with *β* in the revision. Because the appropriate way to calculate degrees of freedom in multilevel models is still being debated (e.g.,https://stat.ethz.ch/pipermail/r-help/2006-May/094765.html), we decided to report *t*values and confidence intervals, as these provide an accurate, precise, and interpretable metric of statistical reliability. The corresponding statistics for the test phase are reported in Table 1. We now state this approach explicitly in the Methods (pg. 11).

Results: Action Representations during Early Encoding and Preparation Phase, 3rd paragraphThe authors found no significant effect of test probability on representations. For factors stimulus and rule that seems to reflect that there was no prioritization. However, for conjunctions, it is difficult to draw such a conclusion given the poor signal-to-noise ratio.

We agree that our use of b-values was confusing in the original submission. To clarify, *b* denoted coefficients of the fixed effects of the multilevel models. To avoid confusion, we replaced *b* with *β* in the revision. Because the appropriate way to calculate degrees of freedom in multilevel models is still being debated (e.g.,https://stat.ethz.ch/pipermail/r-help/2006-May/094765.html), we decided to report *t*values and confidence intervals, as these provide an accurate, precise, and interpretable metric of statistical reliability. The corresponding statistics for the test phase are reported in Table 1. We now state this approach explicitly in the Methods (pg. 11).

Results: Output gating of Action Representations During the Test Phase, 2nd paragraphThe authors state: "Specifically, high priority items were more active than low priority items when they were cued, but the reverse was the case when low priority items were cued."I'm having difficulties seeing the mentioned reverse effect. For stimulus representations it seems to not occur. For conjunction and response representations the reverse effect is somewhat transient.

The Reviewer is correct that the reversal effect (Figure 4) was only significant for the conjunctive representations and response representations and not the stimulus representation. The gray bars on the Figure 4 on right side of each row of the panel directly test the interaction (i.e., reversal) effect. The detected effects were indeed transient, in particular for conjunctive representations, yet we believe the effects are important and not to be dismissed. First, although the effects are transient, they are significant based on the cluster-based permutation test, and they last at least 250ms. Further, as this is the first study to consider attention to the conjunctive representation, we have no reason to rule out that control over this representation may indeed occur in a transient manner. Finally, as noted below, we withhold from overinterpreting the null effects in other portions of the test interval, where there was not a significant difference.

We edited the descriptions to clarify the confusions.

Also, in Table 1 it seems surprising that the t-value for stimulus (low priority) /low priority tested is as large as 2.19 while in Figure 5 the corresponding curve undulates around zero. Perhaps I misunderstood how Figure 5 and Table 1 relates to one another. Please clarify.

Table 1 summarizes the statistics that test whether the decoded RSA scores averaged over the early encoding period and maintenance period significantly differ from 0 (i.e., the dependent variables are RSA scores). Figure 5 summarizes models testing how RTs/Errors are modulated by the priority of actions and the test type (i.e., dependent variables are RTs for the bottom panels or errors for the top panels). Thus, we believe the Reviewer is asking about the consistency between Table 1 and Figure 4 (instead of Figure 5). If so, thank you for pointing out this oversight, we discovered that some of the statistics reported in the Table 1, including the result of the low priority stimulus in lowpriority tested condition, were incorrect. We fixed the mistakes (and checked all other reported statistics) and also report the confidence intervals to be consistent to the main text. None of these updates substantially changed the conclusion except that lowpriority stimulus representations in the low-priority tested condition is no longer statistically significantly detected, which is consistent with Figure 4. Again, we thank the Reviewer for calling our attention to this issue.

I'm not sure whether the absence of correlations between high- and low-priority conjunctions is necessarily indicating independent representations that don't compete with one another. Also, wouldn't independent representation contradict what is said in the previous paragraph that high-priority plans interfere with low-priority plans?

We thank the Reviewer for prompting us to clarify this point. We did not intend to claim that the absence of correlations between high- and low-priority conjunctions is positive evidence of independence. Rather, we intended to report that there is no positive evidence for a strong interference in encoding and maintenance of two conjunctions. Furthermore, this absence of correlations is not contradictory to the reported interference effect during selection because encoded information could introduce competitive computations in downstream processes (e.g., gating of working memory information). We have further clarified this point in the result (pg. 14).

Furthermore, the interaction for the conjunction representation arises later than the representation for the response. This seems to be non-intuitive given that the response, in the end, is what is behaviorally required. Please elaborate.

This comment was also noted by the Editor and other reviewers, so we reproduce our response to this point here. Thank you for raising this important point. First, we fixed reported methodological inconsistencies such as the cluster (*P-*value and *cluster*-forming threshold). Further, we fully agree that the difference in the time course for the response and conjunctive representations in the low priority, tested condition is unexpected and would complicate the perspective that the conjunctive representation contributes to efficient response selection. However, additional analysis indicates that this apparent pattern in the stimulus locked result is misleading and there is a more parsimonious explanation. First, we wish to caution that the data are relatively noisy and likely are influenced by different frequency bands for different features. Thus, fine-grained temporal differences should be interpreted with caution in the absence of positive statistical evidence of an interaction over time. Indeed, though Figure 4 in the original submission shows a quantitative difference in timing of the interaction effect (priority by test type) across conjunctive representation and response representation, the direct test of this four way interaction [priority x test type x representation type (conjunction vs. response), x time interval (1500 ms to 1850 ms vs. 1850 to 2100 ms)] is not significant, (*t*(1,23) = 1.65, *β* = .058, 95% CI [-.012.015]). The same analysis using responsealigned data is also not significant, (*t*(1,23) = -1.24, *β* = -.046, 95% CI [-.128.028]). These observations were not dependent on the choice of time interval, as other time intervals were also not significant. Therefore, we do not have strong evidence that this is a true timing difference between these conditions and believe this is likely driven by noise.

Further, we believe the apparent late emergence of difference in two conjunctions when the low priority action is tested is more likely due to a slow decline in the strength of the untested high priority conjunction rather than a late emergence of the low priority conjunction. This pattern is clearer when the traces are aligned to the response. The tested low priority conjunction emerges early and is sustained when it is the tested action and declines when it is untested (-226 ms to 86 ms relative to the response onset, cluster-forming threshold, *p* <.05). These changes eventually resulted in a significant difference in strength between the tested versus untested low priority conjunctions just prior to the commission of the response (Figure 1, the panel on right column of the middle row, the black bars at the top of panel). Importantly, the high priority conjunction also remains active in its untested condition and declines later than the untested low priority conjunction does. Indeed, the untested high priority conjunction does not decline significantly relative to trials when it is tested until after the response is emitted (Figure 1, the panel on right column of the middle row, the red bars at the top of panel). This results in a late emerging interaction effect of the priority and test type, but this is not due to a late emerging low priority conjunctive representation.

In summary, we do not have statistical evidence of a time by effect interaction that allows us to draw strong inferences about timing. Nonetheless, even the patterns we observe are inconsistent with a late emerging low priority conjunctive representation. And if anything, they support a late decline in the untested high priority conjunctive representation. This pattern of the result of the high priority conjunction being sustained until late, even when it is untested, is also notable in light of our observation that the strength of the high priority conjunctive representation interferes behavior when the low priority item is tested, but not vice versa. We now address this point about the timing directly in the results (pg. 15-16) and the discussion (pg. 21), and we include the response locked results in the main text along with the stimulus locked result including exploratory analyses reported here.

Results: Figure 5 and corresponding textPlease explain why "facilitatory effects were diminished or reversed into interference when the low priority item was tested as a function of the strength of the high priority representation." From what I gather there is a more negative correlation between reaction times and errors for low-priority representations when they are actually tested, or more generally speaking: there seems to be a more negative correlation for low- as well as for high-priority representations when these representations are tested. So then, I don't understand why the correlations for high- and low-priority representations reflect an asymmetry or interference.

The reviewer’s interpretation is correct: we observed a more negative correlation for low- as well as for high-priority representations when these representations are tested. To be more specific, in our paradigm, we can track both high- and low-priority action features (stimuli, conjunctions, responses) independently. This allows us to test how different levels of priority influence selection of an action as a function of test type (tested high priority or low priority action). In this case, a negative correlation of the RSA score for these representations with RT and error indicate a facilitatory effect and a positive correlation indicates an interference effect. As expected based on previously published results (e.g., Kikumoto and Mayr, 2020), we observed that higher RSA scores (i.e., better decoding) of representations of the tested action correlated with facilitated selection. However, the effect of higher RSA scores was *diminished or reversed into interference* when the corresponding action was not tested. For example, having higher RSA scores of the high-priority conjunction and high-priority response was positively correlated with error independently when the low-priority action was tested. A similar trend was observed for low priority action features, but they showed much weaker effects in general (i.e., asymmetric influences on behavior between high and low priority action representations). We agree this point is confusing, but it is an important observation. So, we have added additional exposition in the main text to clarify this analysis and the interpretation of the multilevel modeling results.

Also, it is stated that there is no correlation for low-priority representations but, once again, I'm not sure whether any conclusions can be derived here given the poor signal-to-noise ratio.

Thank you for raising an important point. We have taken care in the revision to state that we find evidence of an interference effect for the high-priority item and do not find evidence for such an effect from the low-priority item. Thus, we do not intend to infer that no such effect could exist. Further, although it is not our intention to draw a strong conclusion from the null effect (i.e., no correlations), we performed an exploratory analysis in which we tested the correlation in trials where we observed strong evidence of both conjunctions. Specifically, we binned trials into half within each time point and individual subject and performed the multi-level model analysis using trials where both high and low priority conjunctions were above their medians. Thus, we selected trials in such a way that they are independent of the effect we are testing. Author response image 1 shows the coefficient of associated with low-priority conjunction predicting high-priority conjunction (uncorrected). Even when we focus on trials where both conjunctions are detected (i.e., a high signal-to-noise ratio), we observed no tradeoff. Again, we cannot draw strong conclusions based on the null result of this exploratory analysis. Yet, we can rule out some causes of no correlation between high and low priority conjunctions such as the poor signal-to-noise ratio of the low priority conjunctions. We have further clarified this point in the result (pg. 14).

**Author response image 1. sa2fig1:** Trial-to-trial variability between high and low priority conjunctions, using above median trials. The coefficients of the multilevel regression model predicting the variability in trial-to-trial highpriority conjunction by low-priority conjunction.

In addition, the representations in the left vs. right column should only differ after 1500ms. I see quite a bit of difference that might be important to account for in any statistical analysis.

To clarify, the result plotted in Figure 5 is using RSA scores averaged during the test (i.e., selection) period, which ranges from 1500 ms to 2200 ms.

Reviewer #2 (Recommendations for the authors):In essence, the authors should be more careful in interpreting their results. While I think that a clever experimental design was used, I also think that quite a lot of processing steps are necessary to reach the given conclusions. Therefore, the authors need to pay more attention to justifying all of these processing steps (e.g., see different p-thresholds).

We agree with the reviewer and we added the results of additional control analyses as discussed above and alternative interpretations (e.g., temporal order of the interaction effect between the priority of actions and the test type). Also, we agree that it is important to justify how different processing steps are necessary to characterize action representations. We updated a clarification and explained in the method section.

Furthermore, I find the results regarding the interaction of test-type and priority factors unconvincing in their current form. The time course of these effects does not seem very plausible to me. Here, the authors should highlight inconsistencies more clearly in the discussion of the results.It is very important to emphasize more clearly on what basis the authors determined the individual RSA model vectors (stimulus, rule, conjunction, response) and how independence could be guaranteed here. This is mentioned briefly in the methods section. For readers who are not particularly familiar with the application of RSA on the basis of decoding result patterns (and I would honestly count myself among them), this part is unfortunately not particularly easy to understand. More background information on the combination of these methods would be necessary here (especially regarding the RSA model vectors and the multilevel models).

We thank the reviewer for highlighting where these methods can be clarified. We have added the theoretical background that motivated us to include the RSA model of conjunctive representation. Specifically, several theories of cognitive control proposed that over the course of action planning, the formation of such an integrated representation is essential for an action to be executed (Frings et al., 2020; Henson et al., 2014; Hommel, 2004; Logan, 1989). Recent studies that applied a decoding analysis to the distributed pattern of EEG activity revealed that humans form such conjunctive representations while preparing actions. The temporal trajectories of conjunctive representations were dissociable from other action representations, and its trial-to-trial fluctuation strongly correlated with efficient action control. To characterize the conjunctive representations, we used a time resolved representational similarity analysis (RSA) which included the RSA models of the rule, stimuli and responses of two actions. In addition, the conjunction RSA model, which tests whether a specific instance of action is distinct from other S-R mappings, was included to assess the variance explained by a unique prediction of the pattern of similarity over and above the patterns expected by other constituent action features.

More details are required in the Methods section:What were the performance-based incentives about?

Performance-based incentives were used to encourage subjects to prepare actions as early as possible to produce accurate and fast responses. Because experiment sessions were long (> 2.5 hours), we used the performance-based incentives to keep subjects engaged with the task. [Detail the performance-based incentive procedure.]

The adaptive response interval was decreased after each correct trial and additionally decreased after five correct trials? Or was it increased after incorrect trials and decreased after five correct trials? The respective sentence seems quite confusing to me.

The adaptive response cutoff interval was adaptively adjusted trial-to-trial by increasing the interval after a single incorrect trials while decreasing the interval after five consecutive correct trials. [Provide the rationale for this asymmetry] We have revised the main text to clarify this point.

How many epochs were excluded from the analysis? What was the number of epochs remaining for each experimental condition?

Using the reported rejection criteria, we excluded 268 trials of trials in each subject on average. We retained an average of 1100 trials for the condition where the high priority action was tested, and 472 trials for the low priority action test condition. Please note that to account for the difference in the number of trials for test type, we used all nonrejected trials irrespective of conditions to train decoding models then analyzed their performance separately by conditions. The number of retained trials for each stimulus response mapping of high and low priority actions are summarized in the Author response image 2. We added a clarification and explained this limitation in the method section**.**

**Author response image 2. sa2fig2:** The number of retained trials of each S-R mapping condition. The box plot shows the count of number of trials retained after artifact rejection. The number label of each S-R mapping corresponds to Figure 1C in the main text.

Reviewer #3 (Recommendations for the authors):1. "Conjunctive representations were significantly modulated by the selection demand (i.e., cued as an output or not) dependent on their priority (Figure 4; see Table 2 for the main effects and Supplementary Figure 1 for the results using response-aligned data)". However, when looking at Figure 4, it seems that in the "tested low priority" condition, the conjunction representation strength of low priority action becomes higher than high priority action (middle row) only after the representation of the low priority response passes the high priority response (bottom row). The same pattern can also be observed in Figure 4 – supplement 1, in which t-values are time-locked to responses. If the conjunction is used to produce the response, its "cross-over" should precede the "cross-over" of responses. Given this analysis is key to the authors' conclusion that the conjunction representation is subject to output gating, the authors need to address this order issue by looking into a formal statistical test of the order of the two cross-over effects (or the order of the beginning of the two priority x test interactions) and further discussing the functional importance of the cross-over of the conjunction (e.g., whether the change is related to performance in the current trial or post-performance monitoring/adjustment).

This comment was also noted by the Editor and other reviewers, so we reproduce our response to this point here. Thank you for raising this important point. First, we fixed reported methodological inconsistencies such as the cluster (*P-*value and *cluster*-forming threshold). Further, we fully agree that the difference in the time course for the response and conjunctive representations in the low priority, tested condition is unexpected and would complicate the perspective that the conjunctive representation contributes to efficient response selection. However, additional analysis indicates that this apparent pattern in the stimulus locked result is misleading and there is a more parsimonious explanation. First, we wish to caution that the data are relatively noisy and likely are influenced by different frequency bands for different features. Thus, fine-grained temporal differences should be interpreted with caution in the absence of positive statistical evidence of an interaction over time. Indeed, though Figure 4 in the original submission shows a quantitative difference in timing of the interaction effect (priority by test type) across conjunctive representation and response representation, the direct test of this four way interaction [priority x test type x representation type (conjunction vs. response), x time interval (1500 ms to 1850 ms vs. 1850 to 2100 ms)] is not significant, (*t*(1,23) = 1.65, *β* = .058, 95% CI [-.012.015]). The same analysis using response aligned data is also not significant, (*t*(1,23) = -1.24, *β* = -.046, 95% CI [-.128.028]). These observations were not dependent on the choice of time interval, as other time intervals were also not significant. Therefore, we do not have strong evidence that this is a true timing difference between these conditions and believe this is likely driven by noise.

Further, we believe the apparent late emergence of difference in two conjunctions when the low priority action is tested is more likely due to a slow decline in the strength of the untested high priority conjunction rather than a late emergence of the low priority conjunction. This pattern is clearer when the traces are aligned to the response. The tested low priority conjunction emerges early and is sustained when it is the tested action and declines when it is untested (-226 ms to 86 ms relative to the response onset, cluster-forming threshold, *p* <.05). These changes eventually resulted in a significant difference in strength between the tested versus untested low priority conjunctions just prior to the commission of the response (Figure 1, the panel on right column of the middle row, the black bars at the top of panel). Importantly, the high priority conjunction also remains active in its untested condition and declines later than the untested low priority conjunction does. Indeed, the untested high priority conjunction does not decline significantly relative to trials when it is tested until after the response is emitted (Figure 1, the panel on right column of the middle row, the red bars at the top of panel). This results in a late emerging interaction effect of the priority and test type, but this is not due to a late emerging low priority conjunctive representation.

In summary, we do not have statistical evidence of a time by effect interaction that allows us to draw strong inferences about timing. Nonetheless, even the patterns we observe are inconsistent with a late emerging low priority conjunctive representation. And if anything, they support a late decline in the untested high priority conjunctive representation. This pattern of the result of the high priority conjunction being sustained until late, even when it is untested, is also notable in light of our observation that the strength of the high priority conjunctive representation interferes behavior when the low priority item is tested, but not vice versa. We now address this point about the timing directly in the results (pg. 15-16) and the discussion (pg. 21), and we include the response locked results in the main text along with the stimulus locked result including exploratory analyses reported here.

2. Is it possible to explore the key channels and/or frequency bands that are important in decoding the conjunction representations?

We thank the reviewer for encouraging us to conduct these additional exploratory analyses, as we agree they can be informative. We performed searchlight RSA analyses to subsample training data in the frequency and channel space. Specifically, to investigate the frequency space, we replicated the combination of decoding and RSA analyses separately for the δ (1-3 *Hz*), theta (4-7 *Hz*), α (8-12 *Hz*), β (13-30 *Hz*) and γ (31-35 *Hz*) frequency bands. For the channel (electrodes) space, we defined ROIs as a group of electrodes of F3-C3-T3, F4-C4-T4, Fz-Cz, OL_O1, OR-O2, P3-PO3-T5, P4-O4-T6, and Pz-POz. To reduce the influence of temporal smearing, which could differ across frequency-bands (2), we averaged data over a priori selected time intervals (i.e., encoding, preparation, and test phase) prior to the training of decoders. Note that by averaging signals, we focus on the informative EEG patterns that are temporally stationary. Other steps in the analysis were identical to the one described in *Representational Similarity Analysis* section.

Overall, participants exhibited considerable variability in the frequency-bands and the position of electrodes in which specific information of prepared actions was encoded. We did not find any systematic differences among individuals in the channels differentially contributing to code action representations (not shown). In the frequency domain, we found that slow rhythmic activities in δ and theta-band lead to better decoding performance in general, where such an effect was more pronounced for decoding of the stimuli and rule. In the revised manuscript, we note these control analyses in the supplemental materials.

**Author response image 3. sa2fig3:** Decoded representations of two actions using frequency-specific scalptopography. The RSA scores of individual subjects using EEG signals in specific frequency ranges (1-3 Hz for the δ-band, 4-7 *Hz* for the theta-band, 8-12 *Hz* for the α- band, 13-30 for the β-band, 31-35 *Hz* for the γ-band, and 1-35 *Hz* for all). EEG signals were averaged over encoding (0 to 750 ms), maintenance (750 to 1500 ms) selection (1500 to 220 ms) time intervals before decoding analysis.